# Sun photometer retrievals of Saharan dust properties over Barbados during SALTRACE

Carlos Toledano[1], Benjamín Torres[2], Cristian Velasco-Merino[1], Dietrich Althausen[3], Silke Groß[4], Matthias Wiegner[5], Bernadett Weinzierl[6], Josef Gasteiger[6], Albert Ansmann[3], Ramiro González[1], David Mateos[1], David Farrel[7], Thomas Müller[3], Moritz Haarig[3], and Victoria E. Cachorro[1]

[1]Group of Atmospheric Optics, University of Valladolid, Valladolid, Spain
[2]Laboratory of Atmospheric Optics, University of Lille, Villeneuve d'Ascq, France
[3]Leibniz Institute for Tropospheric Research, Leipzig, Germany
[4]German Aerospace Center, Institute of Atmospheric Physics, Oberpfaffenhofen, Germany
[5]Meteorological Institute, Ludwig Maximilians University, Munich, Germany
[6]Aerosol Physics and Environmental Physics, University of Vienna, Vienna, Austria
[7]Caribbean Institute for Meteorology and Hydrology, Bridgetown, Barbados

**Correspondence:** Carlos Toledano
(toledano@goa.uva.es)

**Abstract.** The Saharan Aerosol Long-range Transport and Aerosol-Cloud-Interaction Experiment (SALTRACE) was devoted to the investigation of Saharan dust properties over the Caribbean. The campaign took place in June-July 2013. A wide set of ground-based and airborne aerosol instrumentation was deployed at Barbados island for a comprehensive experiment. Several sun photometers performed measurements during this campaign: two AERONET Cimel sun photometers and the Sun and Sky Automatic Radiometer (SSARA). The sun photometers were co-located with the ground-based multi-wavelength lidars BERTHA and POLIS. Aerosol properties derived from direct sun and sky radiance observations are analyzed, and a comparison with the co-located lidar and in-situ data is provided. The time series of aerosol optical depth allows identifying successive dust events with short periods in between in which the marine background conditions were observed. Moderate aerosol optical depth in the range 0.3 to 0.6 was found during the dust periods. The sun photometer infrared channel at 1640nm wavelength was used in the retrieval to investigate possible improvements to aerosol size retrievals and expected larger sensitivity to coarse particles. The comparison between column (AOD) and surface (dust concentration) data demonstrates the connection between the Saharan Air Layer and the boundary layer in the Caribbean region, as it is shown by the synchronized detection of the successive dust events in both data sets. However the differences of size distributions derived from sun photometer data and in-situ observations reveal the difficulties to carry out a column closure study.

# 1 Introduction

Mineral dust is a major contributor to natural aerosol particles. The Sahara desert is the main source of natural dust in the northern hemisphere (Goudie and Middleton, 2001). Mineral dust has important effects on climate due to its interaction with solar radiation (Liao and Seinfeld, 1998) and its contribution to modify cloud properties and processes (Tang et al., 2016). It is largely known that dust originated in the Sahara desert is transported across the Atlantic Ocean to the Caribbean (Prospero and Carlson, 1972; Prospero, 1999). Thus, Saharan dust affects vast areas although the modification of properties and effects along this transport over the Atlantic ocean are still not well understood.

The understanding of all the complex phenomena taking place in the Saharan Air Layer (Carlson and Prospero, 1972) can only be tackled with a combination of long-term observations of key variables using ground-based, airborne and satellite techniques, and comprehensive field experiments that include multiple state-of-the-art instrumentation and a synergistic analysis including the necessary link to the modeling efforts (Tegen, 2003; Heinold et al., 2011; Gasteiger et al., 2017). A list of field experiments undertaken in the last decade aiming at the characterization of mineral dust is provided by Weinzierl et al. (2017).

Sun photometer observations within the Aerosol Robotic Network (AERONET, Holben et al., 1998) provide long-term observation of the atmospheric aerosol at global scale. In the Caribbean, AERONET observations have been carried out since 1996 at Barbados. A number of sites have incorporated to the program in the last decade. A list of sites and available measurements is provided by Velasco-Merino et al. (2018) as well as on the AERONET website. Furthermore, ground-based in situ observations of Saharan dust are carried at Barbados since 50 years. The Barbados dust record (Prospero et al., 2014) started in 1965 and is the longest existing record of ground-based dust measurements. Both AERONET data and ground-based dust concentrations allow us evaluating the SALTRACE data in a long-term context. These data indicate that dust conditions during the SALTRACE observation period in summer 2013 were 'typical' (Weinzierl et al., 2017).

The Saharan Aerosol Long-Range Transport and Aerosol-Cloud-Interaction Experiment (SALTRACE, Weinzierl et al., 2017) took place in Barbados in 2013-14. The first observation period of SALTRACE was conducted in the Caribbean region in June-July 2013. Two other observation periods were carried out in spring and summer 2014. The aim of this experiment was to collect a wide set of ground-based and airborne-based measurements of long-range transported Saharan dust, in order to provide a unique dataset comprising in-situ and remote sensing derived aerosol and cloud variables. These data will allow analyzing to what extent aged mineral dust changes its properties during transport and how these particles play a role in cloud processes in the Caribbean area.

This experiment is closely connected with the Saharan Mineral Dust Experiment (SAMUM, Heintzenberg, 2009; Ansmann et al., 2011), that was carried out in the vicinity of the Saharan desert and at the beginning of the long-range transport over the Atlantic, in the Cape Verde islands. In fact most of the instrumentation (ground-based and airborne) was consecutively deployed at Morocco (SAMUM-1), Cape Verde (SAMUM-2) and finally Barbados (SALTRACE). During this time, instruments and retrievals have been continuously improved in order to reduce experimental uncertainties and provide the best possible data. Some examples of these efforts in the field of remote sensing are the use of complex shapes in the modeling of particles and inversion of lidar data (Gasteiger et al., 2011) or the improvements in polarization measurements from lidar, with the

development of enhanced calibration schemes (Freudenthaler et al., 2009; Freudenthaler, 2016), enhanced instrumentation and extended spectral range (Gasteiger and Freudenthaler, 2014; Groß et al., 2015; Haarig et al., 2017). The AERONET version-3 processing algorithm (Giles et al., 2019) is another example as it provides enhanced retrieval of basic properties, like the aerosol optical depth, as well as advanced characterization of aerosol optical properties. At the same time, it improves the cloud-screening and quality control algorithms.

The aim of this paper is to analyze the sun photometer observations carried out during SALTRACE, and relate them to the co-located aerosol measurements. Section 2 describes the measurement sites and instrumentation involved in this study. Then the methodology is briefly described in section 3 and the results are presented in section 4, including aerosol optical depth, inversion products and a comparison with in-situ observations.

## 2 Sites and instrumentation

The measurements presented in this paper were collected at two sites, both located in Barbados island. The first one was set at the headquarters of the Caribbean Institute for Meteorology and Hydrology (CIMH), in the western part of Barbados (13°8'55"N, 59°37'29"W, 110m a.s.l.). The AERONET measurements in this location can be found under the site name 'Barbados SALTRACE'. The CIMH was a supersite for SALTRACE, with several remote sensing instruments such as two AERONET sun photometers, the SSARA sun-sky radiometer (Toledano et al., 2009), the POLIS lidar (Groß et al., 2015, 2016) and the BERTHA lidar (Haarig et al., 2017), together with ancillary data from meteorological radiosondes and a hand-held Microtops for ozone measurements. CIMH is located on a small hill, 1.5km away from the coast, in the vicinity of the city of Bridgetown, with about a hundred thousand inhabitants.

The second site was located in Ragged Point (13°9'54"N, 59°25'56"W, 40m a.s.l.), in the eastern coast of Barbados island and 25km away from CIMH. At Ragged Point a set of in-situ instrumentation was deployed in a measurement tower, including samplers and optical instrumentation for the derivation of dust optical, chemical and micro-physical properties at ground level (Kristensen et al., 2016; Kandler et al., 2018). This is the regular measurement site of the 50-year Barbados dust record mentioned above. Given the privileged location of Barbados as the easternmost island of the Caribbean, it makes possible to measure undisturbed African dust after its long-range transport over the Atlantic. An AERONET sun photometer is also operated on a continuous basis at Ragged Point. A comprehensive list of instruments in the CIMH and Ragged Point sites during SALTRACE can be found in the supplemental material of Weinzierl et al. (2017).

The sun photometers had different features and were operated at different configurations in order to cover as many aspects as possible. Two AERONET Cimel sun photometers (Holben et al., 1998) were operated at CIMH during SALTRACE. Table 1 summarizes the spectral ranges, measurement types and frequencies performed by each sun photometer. These instruments perform direct Sun observations at 9 spectral channels in the range 340-1640nm plus sky radiance in six channels (440, 500, 675, 870, 1020 and 1640nm) in the principal plane and almucantar geometries. In order to optimize the observations, one of the Cimel was set to use high-frequency (3 minute sampling) for aerosol optical depth (AOD) observations; and the other one,

equipped with polarization capabilities, was focused on the measurement of sky radiances, especially sky polarization in the principal plane geometry.

The SSARA sun-sky radiometer performs continuous direct sun observations at 12 spectral channels ranging from 340 to 1550 nm. Sky radiances in the solar almucantar and principal plane geometries are measured every 30 minutes at 440, 780 and 1020nm. For this campaign new polarization capabilities were added to SSARA (therefore renamed 'SSARA-P'), in order to derive the degree of linear polarization (DOLP) of the sky light at 500nm wavelength.

The ground-based in-situ aerosol measurements included in the present study were carried out at Ragged Point site. The ambient aerosol was sampled through a PM10 inlet located at the top of a 17 m high tower, ca. 50 m a.s.l. The particle number size distributions were measured every 14 min. with an Aerodynamic Particle Sizer (APS-3321, TSI) covering the size range 0.5-10 $\mu m$, and a mobility particle size spectrometer (MPSS, TROPOS-REF-3) measuring the size range 0.01-0.8 $\mu m$ (Kristensen et al., 2016). As for the lidar instrumentation, POLIS is a six-channel lidar system measuring Raman (387 and 607nm) and elastic (355 and 532 nm, cross- and parallel-polarized) backscattered signals for aerosol property profiling (Groß et al., 2015). BERTHA is a multiwavelength polarization/Raman lidar upgraded for this campaign with new channels to allow the observations of dust linear depolarization ratios at 355, 532, and 1064 nm (Haarig et al., 2017). For further information concerning the lidar and in-situ data used in this study, we refer the reader to the corresponding publications in the SALTRACE special issue (Groß et al., 2015, 2016; Kristensen et al., 2016; Haarig et al., 2017, see complete list at https://www.atmos-chem-phys.net/special_issue382.html), as well as previous studies using this instrumentation (Wiegner et al., 2011; Tesche et al., 2011; Müller et al., 2011; Schladitz et al., 2011).

## 3 Methodology

All sun photometer observations were made following the AERONET protocols , i.e. high frequency (see table 1) direct sun observations to derive spectral aerosol optical depth, and sky radiance measurements every hour in the almucantar and principal plane geometries.

All Cimel instruments involved in the campaign were calibrated within AERONET procedures (Holben et al., 1998), therefore the AOD absolute uncertainty is 0.01-0.02 (larger for shorter wavelengths), and sky radiance uncertainty is 5%. Similar uncertainty is found for SSARA-P (Toledano et al., 2011). Its direct Sun channels for AOD were calibrated with the Langley plot method at the Environmental Research Station Schneefernerhaus (2650m a.s.l., at the Zugspitze mountain, Germany); radiance and polarization channels were calibrated at the University of Lille (accessed thanks to the ACTRIS-2 project) using the AERONET-Europe reference integrating sphere and polarization box (Li et al., 2018).

The data processing is also standardized. For aerosol optical depth we have used cloud-screened and quality assured (level 2.0) data of the version 2 processing in the AERONET database. However the differences in AOD with the new version 3 processing (Giles et al., 2019) are minor (below 0.003 in all wavelength channels for any measurement). The use of version 2 AOD is chosen for a better comparison with the previous sun photometer results of the SAMUM campaigns (Toledano et al., 2009, 2011).

For the inversion of the sky radiances, we have self applied the inversion code by Dubovik et al. (2006) to both the almucantar and principal plane geometries, using the four spectral channels that are used in the operational AERONET processing, i.e. 440, 675, 870 and 1020nm wavelengths. We also made an alternative processing by adding the observations at 500 and 1640nm wavelengths, therefore enlarging the observation spectral range in the short wave infrared. The differences provided by this enhanced retrieval will be shown in section 4.2. Neither the principal plane retrievals nor the 6-wavelength retrievals are available in the AERONET database.

In the AERONET operational products, only retrievals with AOD(440nm)>0.4 can be considered as 'Quality Assured'. In our case, we have applied a set of quality criteria to ensure the reliability of our inversion data, which basically are the same conditions of AERONET level 2.0 inversions except for the AOD threshold, i.e. solar zenith angle >50°, minimum number of symmetrical angles and retrieval error between 5% and 8% depending on AOD. We also impose AOD(440nm)>0.2 for the single scattering albedo and complex refractive index (Dubovik et al., 2006; Mallet et al., 2013; Mateos et al., 2014; Burgos et al., 2016; Velasco-Merino et al., 2018). This threshold of AOD(440nm)>0.2 results in an estimated uncertainty of 0.03 for the single scattering albedo, 0.04 for the real part of the refractive index, 50% for the imaginary part and 35% for the volume size distributions (Dubovik et al., 2000).

Due to the frequent presence of cumulus clouds in this area, the cloud screening procedure and the inversion of sky radiances have been carefully checked. Besides the AERONET cloud screening (Smirnov et al., 2000a; Giles et al., 2019), we added manual inspection to the dataset to avoid any cloud contamination. The principal plane radiances cannot be easily screened out for clouds due to the lack of symmetry between branches (as it is the case for the almucantar). Therefore they were manually inspected after a selection of cases in which both the nearby AOD and almucantar observations were cloud free.

## 4 Results

### 4.1 Aerosol optical depth

The aerosol optical depth during the campaign period is shown in Figure 1a. The number of measurement days was 30, with a total number of 940 cloud-free direct Sun observations. The AOD measurements monitor the aerosol content in the atmospheric column. The shaded areas in the figure indicate the five successive dust events that were detected during the SALTRACE experiment at Barbados, with moderate AOD (500nm) up to 0.6. Mean AOD values for the events ranged between 0.22 and 0.42. Dramatic changes can be observed depending on the dust advection (e.g. 12 June). There were also short interruptions in which the marine background aerosol was measured. This general sequence of aerosol events was also identified with atmospheric profiling techniques (Groß et al., 2015; Haarig et al., 2017) as well as the in-situ dust concentration at the ground (Weinzierl et al., 2017; Kandler et al., 2018). No significant differences are found between 'Barbados_Saltrace' and 'Ragged_Point' sites, which is an important result for all comparisons that can be made among in-situ and remote sensing instruments located in these two sites. Moreover, all instruments co-located at CIMH agree within the nominal AOD uncertainty (0.02) for simultaneous measurements. Some minor differences between SSARA and the AERONET Cimels are to be expected in cloudy days (e.g. 17-Jun, 28-Jun). This is due to the different data sampling of the instruments: ('triplets' or 3 measurements

within 1 minute for the Cimels; 2 second sampling for the SSARA), that may yield to non-simultaneous data as well as different results in the cloud-screening process.

The Ångström exponent (AE) depicted in Figure 1b reveals low values during the dust-dominated days in the range 0-0.2, as can be expected for the coarse-dominated mineral aerosol. Even slightly negative AE is observed in some cases. Somewhat higher AE values are observed during the clean (marine-dominated) days. It is also clear from the plot that the agreement among instruments is worse in these pristine days, but this result is to be expected because the uncertainty in AE dramatically increases when AOD is low (Cachorro et al., 2008).

In order to provide the properties of Saharan dust after the transit over the Atlantic, we need to identify the dust-dominated days from the observation period. Given that the Ångström exponent remains low, we simply selected $AOD(500nm) > 0.15$ to separate dust and marine aerosol observations, following the criterion provided by Smirnov et al. (2002) to identify pure marine aerosol. The uppermost rows of table 2 show the statistics for AOD and AE during the dust dominated days of the campaign.

The long deployment of Cimel #440 for more than 1 year at CIMH ('Barbados_Saltrace') allows establishing the marine aerosol background for each month. For its determination, we used percentile 1 of AOD within each month (instead of the minimum value). For June and July this is 0.04 for AOD (500nm). Outside these months, the background was even lower, about 0.02-0.03 for AOD(500nm). This is the kind of atmospheric situation that we observed for very short periods during SALTRACE, as shown in Figure 1a. This background marine aerosol always contributes to the aerosol optical depth observed in the atmospheric column.

In previous works it was shown that the wavelength dependence of the aerosol optical depth does not follow the Ångström power law, especially in the short wave infrared for mineral dust aerosol (Toledano et al., 2011). An example of this feature for SALTRACE data is provided in Figure 2. As can be seen in the plot, the classical fit of spectral AOD to the Ångström formula (Angström, 1961) over the visible range (440-870nm) would largely overestimate the observed AOD at 1640nm wavelength. A second order fit in logarithmic space is needed to properly capture the spectral variation of AOD in the short wave infrared. This approach has been applied to the AOD data in order to provide AOD at $2\mu$m wavelength. This extrapolated AOD($2.022\mu$m) was needed for the calibration of the wind lidar operated during SALTRACE on board the Falcon research aircraft. Thanks to this correction and the co-located POLIS lidar, it was possible to provide aerosol backscatter profiles from the wind lidar (Chouza et al., 2015). Moreover, the consistency between the AOD from sun photometer and the POLIS lidar extinction profiles was demonstrated by Groß et al. (2015).

For the aerosol type identification, we have used the scatter plot of the Ångström exponent vs. aerosol optical depth (500nm), shown in Figure 3. The increasing AOD and decreasing AE pattern for mineral dust is confirmed by SALTRACE data, in a similar way to the SAMUM campaigns also shown in the plot. We also indicate the threshold (AOD=0.15) for separation between marine and dust aerosol predominance for SALTRACE data. The plot indicates that only marine and dust aerosols were present during SALTRACE, with no significant contribution of fine particles, as it was the case during SAMUM-2 in the winter season, resulting in larger AE for fine and coarse particle mixtures. The AE of dust seems to be lower in SAMUM-2 (Cape Verde) and SALTRACE (Barbados) than it was very near the sources in SAMUM-1 (Morocco), which is an unexpected

result. The reason could be the different background aerosol. High AE above 1.0 for low AOD during SAMUM-1 could indicate the presence of fine particles (continental background or anthropogenic pollution) that would result in higher AE than expected for pure dust. Conversely the background aerosol in the island sites is mainly composed of coarse marine particles with associated low AE.

## 4.2 Inversion of sky radiances

The inversion of multi-angle and multi-wavelength diffuse sky radiances, together with spectral AOD, provides volume particle size distribution, complex refractive index (spectral) and fraction of spherical particles (Dubovik and King, 2000; Dubovik et al., 2006). From these basic properties, a set of optical and micro-physical properties of the aerosol particles are obtained: fine mode fraction, effective radius, single scattering albedo, absorption AOD, phase function and asymmetry parameter. In fact, the inversion code can provide column-integrated values for other interesting properties, like lidar ratio, particle linear depolarization ratio, radiation fluxes or radiative forcing. During SALTRACE observation period (30 days) a total number of 54 successful inversions were obtained. The number of data is low compared to the amount of AOD data, both because of the lower measurement frequency and the difficulty to meet the required number of cloud-free observation angles in between the frequent cumulus clouds. Therefore the inversion data are extremely valuable pieces of data, that provide advanced information about aerosol properties that are key for the aerosol radiative effect evaluation. The statistics for some of these parameters are presented in Table 2. It must be noted, however, that a certain contribution of the marine aerosol to the column aerosol properties is always present, therefore values in Table 2 correspond to a mixture of dust (pre-dominant) with some marine aerosol.

One of the above mentioned key properties is the single scattering albedo (SSA) and it spectral dependence. The increase of the single scattering albedo with wavelength is a clear signature for dust (Dubovik et al., 2002). The values observed during SALTRACE are very close to those reported for pure Saharan dust near the sources (Dubovik et al., 2002; Toledano et al., 2009, 2011) and indicate that mineral dust was the predominant aerosol type. The SSA during SALTRACE and its comparison with the SAMUM campaigns is provided in Figure 4. The average was found to be between 0.94 at 440nm and 0.98 at 1020nm for SAMUM-2 and SALTRACE campaigns. The variability is also quite similar. Lower SSA values were observed for SAMUM-1, especially at 440nm. The use of 1640nm wavelength in the retrieval for SALTRACE data shows a decrease in the SSA at this wavelength as compared to 1020nm, in agreement with Müller et al. (2010b).

The volume particle size distributions ($dV/dlnR$) depicted in Figure 5 provide the comparison between SALTRACE and SAMUM campaigns for data with dust predominance. The mean size distribution for SALTRACE (dust cases) has a volume concentration of $0.2\mu m^3/\mu m^2$ and a clear coarse mode predominance: the fine mode fraction of the volume concentrations is 0.09. The effective radii of the fine and coarse mode are $0.15\mu m$ and $1.62\mu m$ respectively. For comparison, the pure dust cases in SAMUM-2 presented effective radii of $0.17\mu m$ and $1.73\mu m$ in the fine and coarse modes, with even more pronounced coarse mode predominance indicated by lower fine mode fraction (0.06). Similarly as described by Velasco-Merino et al. (2018), the long-range transport over the Atlantic produces a decrease in volume concentrations (and AOD), but the change in effective radii of the fine and coarse mode is within uncertainties and therefore not significant. This is agreement with results by Gasteiger et al. (2017) who showed that the fraction of large dust particles does not change much during trans-Atlantic

transport. The fraction of spherical particles was 0.11 in dust cases of SAMUM-2 and 0.12 for SALTRACE (median values). In the short periods of marine background the size distributions of the marine aerosol were retrieved. The average value is given in Figure 5 (dashed line). The volume concentration is clearly lower than for the dust cases. However the coarse mode effective radius is larger for the marine particles ($1.87\mu m$) than it is for dust. This result was also found in other coastal regions (Prats et al., 2011). In the marine cases, the total volume concentration is $0.05\mu m^3/\mu m^2$ and the fraction of spherical particles is 0.90.

As shown in previous figures, we have also compared the inversion products using different wavelength ranges. The 4-wavelength cases include 440, 675, 870 and 1020nm wavelength measurements of AOD and sky radiances. The 6-wavelength cases include the previous 4 channels plus 500 and 1640nm, therefore extending the spectral range considerably[1]. The single scattering albedo (Figure 6a), the real part of the refractive index (Figure 6b) and the fine mode effective radius (Figure 6c), experience limited change if the long wavelength is added. However the coarse mode effective radius (Figure 6d), increases considerably if the 1640nm is used in the inversion. This suggests larger sensitivity to coarse particles, that are otherwise not considered by the inversion code. This fact is also noticeable in Figure 5, in which the coarse mode of the 6-wavelength retrieval is shifted to larger radii as compared to the 4-wavelength retrieval. This is not the case for the fine mode.

Finally we have analyzed two key lidar properties, the lidar (extinction to backscatter) ratio, and the particle linear depolarization ratio. As mentioned above, the inversion of sun photometer data can provide column integrated values for these properties, given that the inversion code computes the full scattering matrix. Therefore the aerosol spheroid model can be used to model lidar observations that are known to be sensitive to non-sphericity of desert dust particles. Figure 7 summarizes the results. For the sun photometer we used the two retrievals (4 and 6 spectral channels), based on inversions of Cimel #789 measurements. The lidar data were obtained with the POLIS lidar at 355 and 532nm (Groß et al., 2015) as well as the BERTHA lidar at 355, 532 and 1064nm (Haarig et al., 2017), and correspond to dust layers.

Figure 7a shows the lidar ratios (LR), which were about 55sr in the POLIS lidar (similar for BERTHA, not shown). Lidar ratios derived from sun photometer are lower, about 50sr. This behavior is opposite to what was found in previous comparisons using SAMUM data (Müller et al., 2010a, 2012), in which LR derived from sun photometer inversion were higher than the values provided by the lidars, especially in the shorter wavelengths. The analysis also shows a large increase for LR at 1640nm observed during SALTRACE; unfortunately lidar data do not provide information that would help to verify the sun photometer retrieval in that spectral region.

Concerning the particle linear depolarization ratio (PLDR), the results are depicted in Figure 7b. This parameter is sensitive to the particle shape, therefore strongly dependent on the spheroid model used for the inversion of sun photometer data. The average values retrieved from sun photometer are 0.25 to 0.27 (440 to 1020nm) with the 4-wavelength retrieval. The 6-wavelength retrieval provides a bit higher values in the shorter wavelengths (0.28 at 440nm) and very similar at 1020nm. More importantly, the wavelength dependence of the PLDR shown by BERTHA lidar is captured by the AERONET 6-wavelength retrieval, and not the 4-wavelength one, even though the PLDR at 1020-1064nm deriven from the sun photometer is in any

---

[1]The key additional channel is 1640nm; the 500nm channel may increase robustness of the results but would not produce the observed changes in the coarse mode retrieval.

case higher than indicated by the lidar uncertainty estimates. The excellent agreement (well within the uncertainty estimates) of the POLIS, BERTHA and sun photometer depolarization ratios at 500-532nm wavelength is also remarkable, in particular as this was not the case for the SAMUM data reported by Müller et al. (2010a). Overall, the spheroid model and the Dubovik inversion seems to provide better results (closer to the lidar measurements) for SALTRACE than it was reported for the
SAMUM campaigns.

## 4.3   Comparison with in-situ observations at Ragged Point

The comparison between remote sensing and in-situ aerosol measurements is a difficult task due to multiple factors. The atmospheric volume that is observed is normally different: it is ground level for routine ground-based monitoring with in situ instrumentation; a profile for lidar that due to overlap issues typically starts some hundreds of meters above the ground; or
the entire atmospheric column for sun photometer retrievals. Only in very well mixed atmospheres with a single predominant aerosol type is it possible to tackle this kind of comparison. Another possibility is to compare lofted layers observed with lidar and aircraft in-situ observations. Other obvious limitations are the presence of inlets, drying elements, etc. needed to introduce the sampled air into the in-situ instruments, as compared to the ambient aerosol measured with remote sensing techniques. Changes in pressure are to be considered in aircraft measurements too. On the other hand, remote sensing data need to be
introduced in inversion algorithms that impose mathematical and physical constraints. Overall, both intensive and extensive aerosol properties need to be compared with extreme caution and it is still an open question to what extent a closure approach is feasible and in-situ and remote sensing observations can be reconciled. Several works already showed these difficulties and reported contradictory results in some cases (Reid et al., 2003; Ryder et al., 2015, and references therein).

Valuable efforts have been also attempted in the context of the SAMUM campaigns (Müller et al., 2010a, b, 2012), in
which very special atmospheric conditions were measured with multiple state-of-the-art instrumentation. The agreements and discrepancies found in those works still remain for many SALTRACE data. For instance, the mean real part of the refractive index (440nm) during SALTRACE retrieved with sun photometer data is 1.47 for mineral dust (Table 2), which is much lower than the value reported from in-situ observations, about 1.55-1.61 (Kandler et al., 2011). The problems encountered to retrieve the correct dust refractive index using a spheroid model were also highlighted by Kemppinen et al. (2015). Conversely, the
single scattering albedo and its spectral dependence (Table 2) are very similar for dust using sun photometer and in-situ optical instruments.

The difficulties to compare in-situ and column (sun photometer) size distributions (e.g. Toledano et al., 2011; Ryder et al., 2015) are still present in SALTRACE data. As an example, we have plotted in Figure 8 the volume size distributions derived from in-situ and sun photometer data for a dust and a no-dust case. We have converted the column size distribution from the
sun photometer (originally in $\mu m^3/\mu m^2$) to concentration in $\mu m^3/cm^3$ assuming that the dust layer is distributed evenly in a layer and using a layer height of 4 km for the dust case and 1.5 km for the marine case (data from the co-located POLIS lidar Groß et al., 2015, Fig. 1). About the shape of the size distributions, the coarse mode discrepancy is strongly produced by the inlet efficiency in the in-situ data (PM10 inlet). However even the fine mode does not present a similar shape, in this case likely produced by the difficulties of the sun photometer inversion to correctly reproduce the fine mode in the presence

of a strongly predominant coarse mode (Torres et al., 2017). In the marine (no-dust) case, the fine mode has similar shape and effective radius, although a strong shift is found in the coarse mode. For the dust case, the effective radius is a factor of 1.8 larger for the sun photometer size distribution.

Nevertheless, some in-situ and column parameters show better correlation, as it is the case for the ground-based dust concentration and the aerosol optical depth. This was already highlighted for SALTRACE (Weinzierl et al., 2017, see Fig. 4) as well as for long-term monthly means at Barbados (Smirnov et al., 2000b). The good correlation and absence of time lag between in-situ and column data indicates that dust transported in lofted layers is, at the same time, mixed down into the boundary layer over Barbados.

## 5   Conclusions

The analysis of the aerosol optical depth revealed a continuous succession of dust events during SALTRACE, only interrupted during short periods in which the marine aerosol background was observed. This is as low as 0.04 for AOD (500nm) during the summer months. In contrast, dust layers resulted in a mean AOD (500nm) of 0.26. The aerosol classification analysis indicates that marine and dust were the only aerosol types observed during the experiment. The wavelength dependence of the aerosol optical depth was investigated in order to provide an estimated value at $2\mu m$ wavelength, needed for the wind lidar data analysis (Chouza et al., 2015). A second order fit was needed to capture the appropriate spectral variation in the short wave infrared.

The sky radiance data were inverted using the Dubovik code for both almucantar and principal plane observation geometries, as well as using two different spectral ranges (440-1020nm and 440-1640nm). Only the coarse mode retrieval (size distribution and corresponding effective radius) changes significantly by enhancing the spectral range, showing a shift toward larger radii. As compared to the retrieved size distributions during SAMUM, the change in effective radii of the fine and coarse modes is within the uncertainties and therefore not significant. The fine mode fraction of the size distribution is on average 0.09, thus indicating a clear coarse mode predominance. During marine (no-dust) cases, the coarse mode effective radius is larger than it is for dust.

Column-integrated values were also investigated for two lidar-relevant properties: lidar ratio and particle linear depolarization ratio. The sun photometer inversion retrieval of these parameters seems to improve with respect to previous comparisons carried out with SAMUM data. However, robust uncertainty estimates for the sun photometer inversion products are needed to corroborate this agreement.

The comparison of sun photometer retrievals with in-situ aerosol properties is still subject of investigation. Even if very good correlation was found between dust concentration at ground level and aerosol optical depth, the comparison of other variables like the particle size distributions requires considerable experimental efforts (e.g. additional aircraft measurements) and is thus still challenging. The analysis of SALTRACE data will continue aiming at a closure approach, i.e. the characterization of the same aerosol parameters with various independent methods.

*Author contributions.* CT and CV led the analysis and manuscript writing. BT performed the sun photometer inversions using Dubovik code. TM provided data and analysis in-situ measurements. AA, MH, SG contributed with lidar data and interpretation. All authors contributed to the scientific analysis, manuscript preparation and revision.

*Acknowledgements.* The authors gratefully acknowledge the effort of AERONET and University of Miami to maintain the Ragged Point site.
5   This research has received funding from the European Union's Horizon 2020 Research and Innovation Programme under grant agreement No 654109 (ACTRIS-2). BW and JG have received funding from the European Research Council (ERC) under the European Union's Horizon 2020 research and innovation programme (grant agreement no. 640458, A-LIFE) and from the Helmholtz Association under Grant VH-NG-606 (Helmholtz-Hochschul-Nachwuchsforschergruppe AerCARE). The funding by MINECO (CTM2015-66742-R) and Junta de Castilla y León (VA100P17) is also acknowledged. We thank Volker Freudenthaler and Meinhard Seefeldner from LMU for all their work in relation
10   to SALTRACE, especially the contribution to the development of SSARA-P photometer.

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

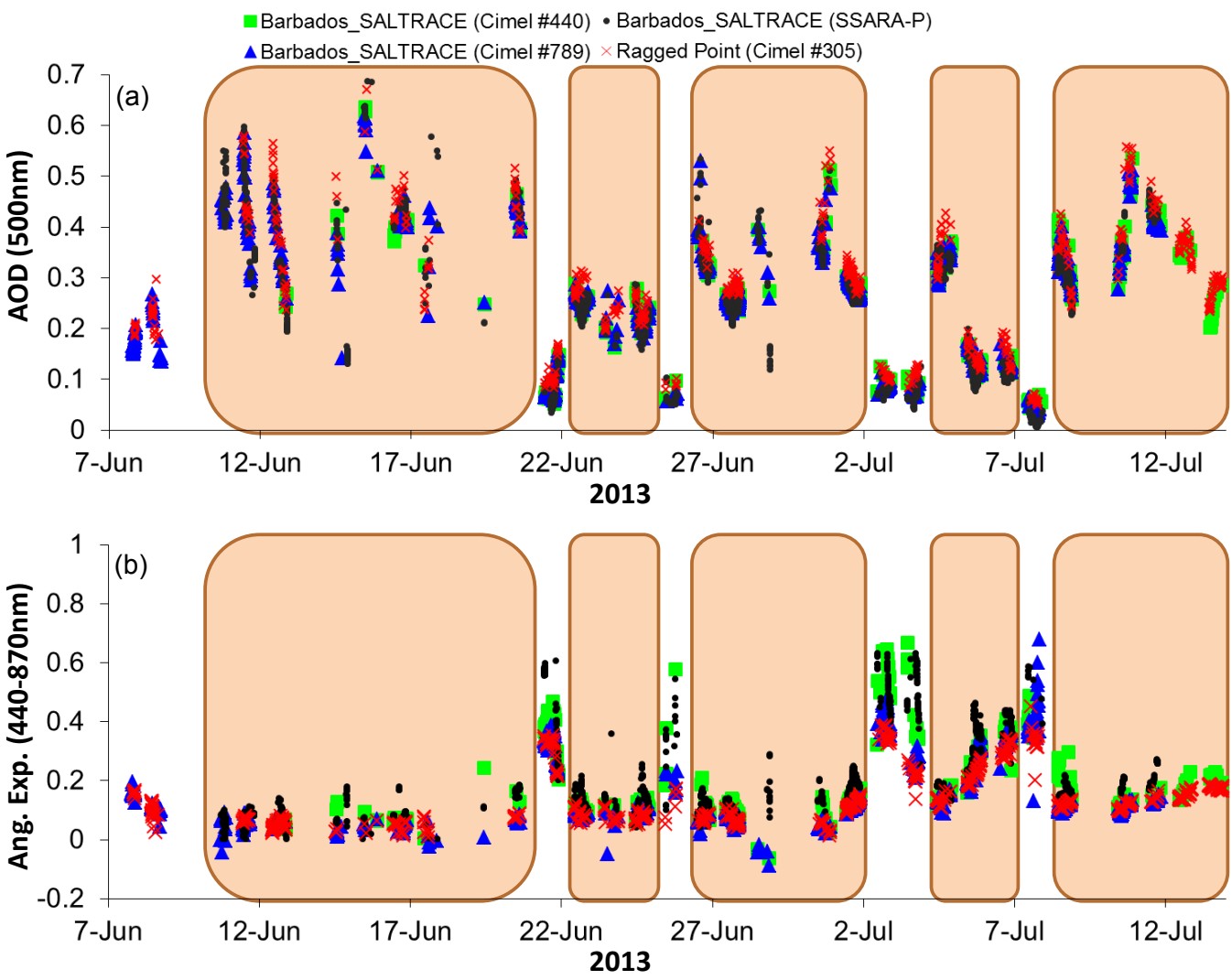

**Figure 1.** Aerosol optical depth (a) and Ångström exponent (b) observations during SALTRACE campaign with all available sun photometers: Cimel #789, Cimel #440 and SSARA-P at CIMH site; and Cimel #389 at Ragged point site. See text and table 1 for details.

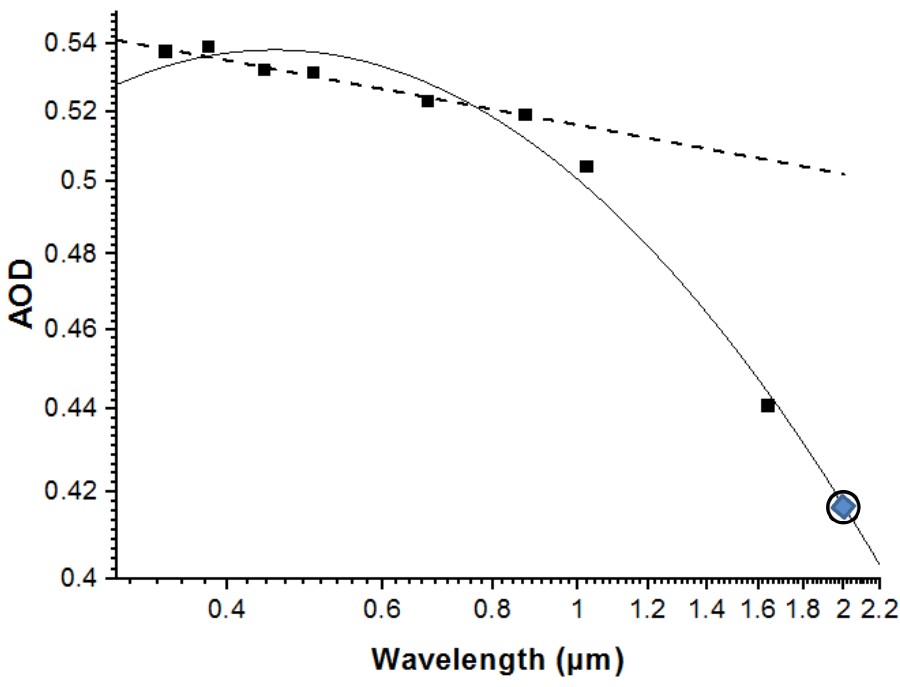

**Figure 2.** Aerosol optical depth as a function of wavelength in log-log scale for 11 June 2013 at 11:12UTC. Solid line indicates the 2nd order fit over the rage 340-1640nm whereas the dashed line indicates the 1st order fit over the range 440-870nm, corresponding to the Ångström formula (Angström, 1961). The extrapolated value of AOD at $2\mu$m wavelength using the 2nd order fit is indicated with a circle.

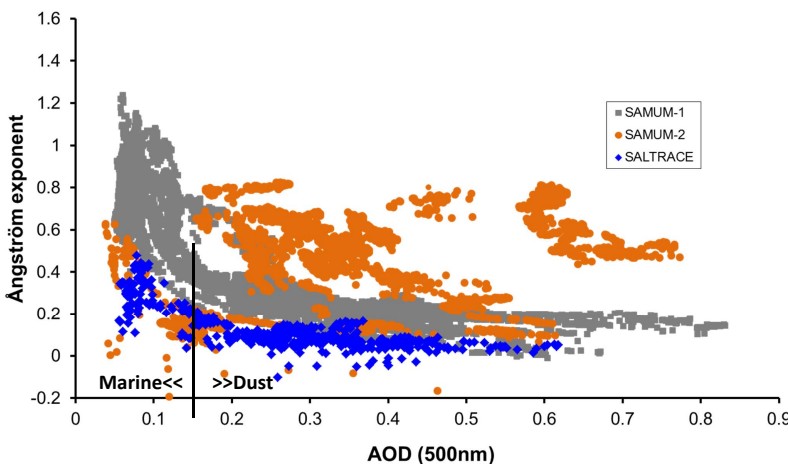

**Figure 3.** Scatter plot of Ångström exponent vs. aerosol optical depth (500nm) for SALTRACE as well as SAMUM-1 and SAMUM-2 campaigns.

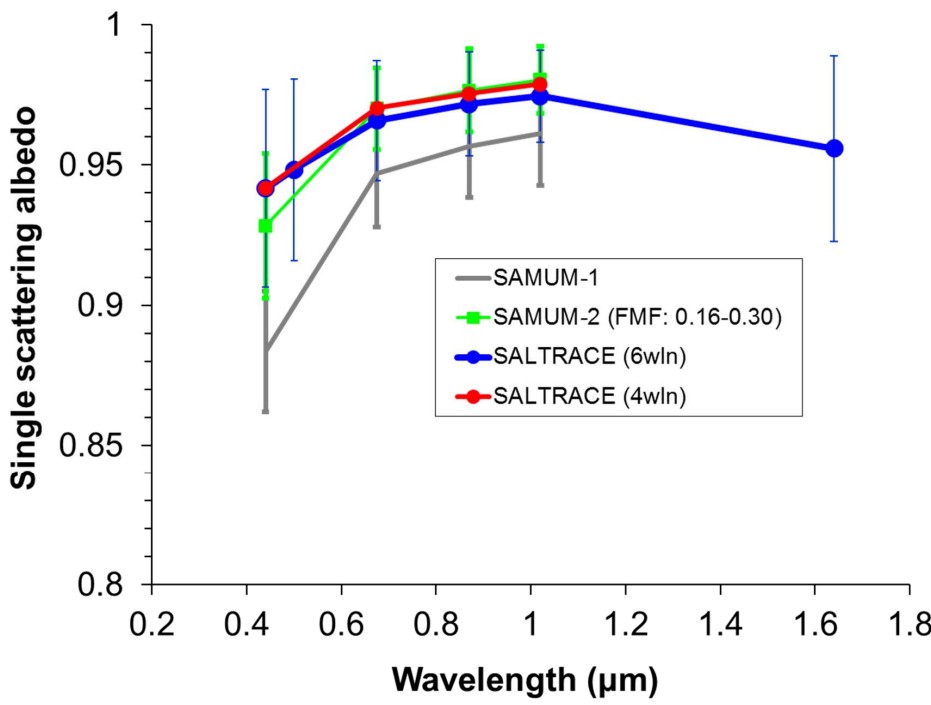

**Figure 4.** Single scattering albedo as a function of wavelength for SALTRACE as well as SAMUM campaigns. Average values for dust cases in each campaign are provided. Bars indicate ±1 standard deviation. For SAMUM-2 only cases with low fine mode fractions (FMF) between 0.16 and 0.30 are considered. For SALTRACE data, retrievals using 6 wavelength (6wln) and 4 wavelength (4wln) channels are shown.

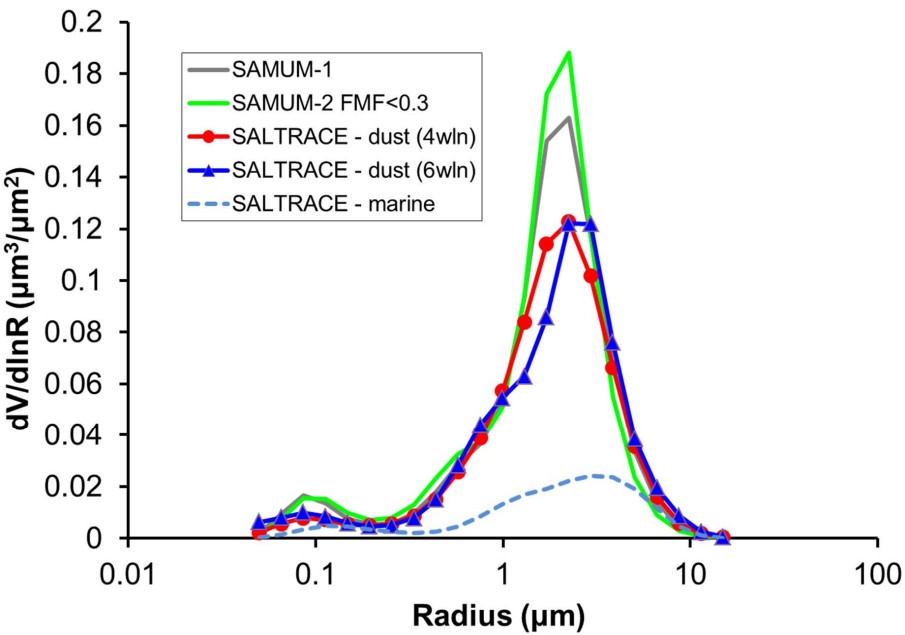

**Figure 5.** Volume particle size distributions (dV/dlnR) for SALTRACE as well as SAMUM campaigns. Average values for dust cases in each campaign are provided, based on inversion of Cimel sun photometer data. For SAMUM-2 only cases with low fine mode fractions (FMF) between 0.16 and 0.30 are considered.

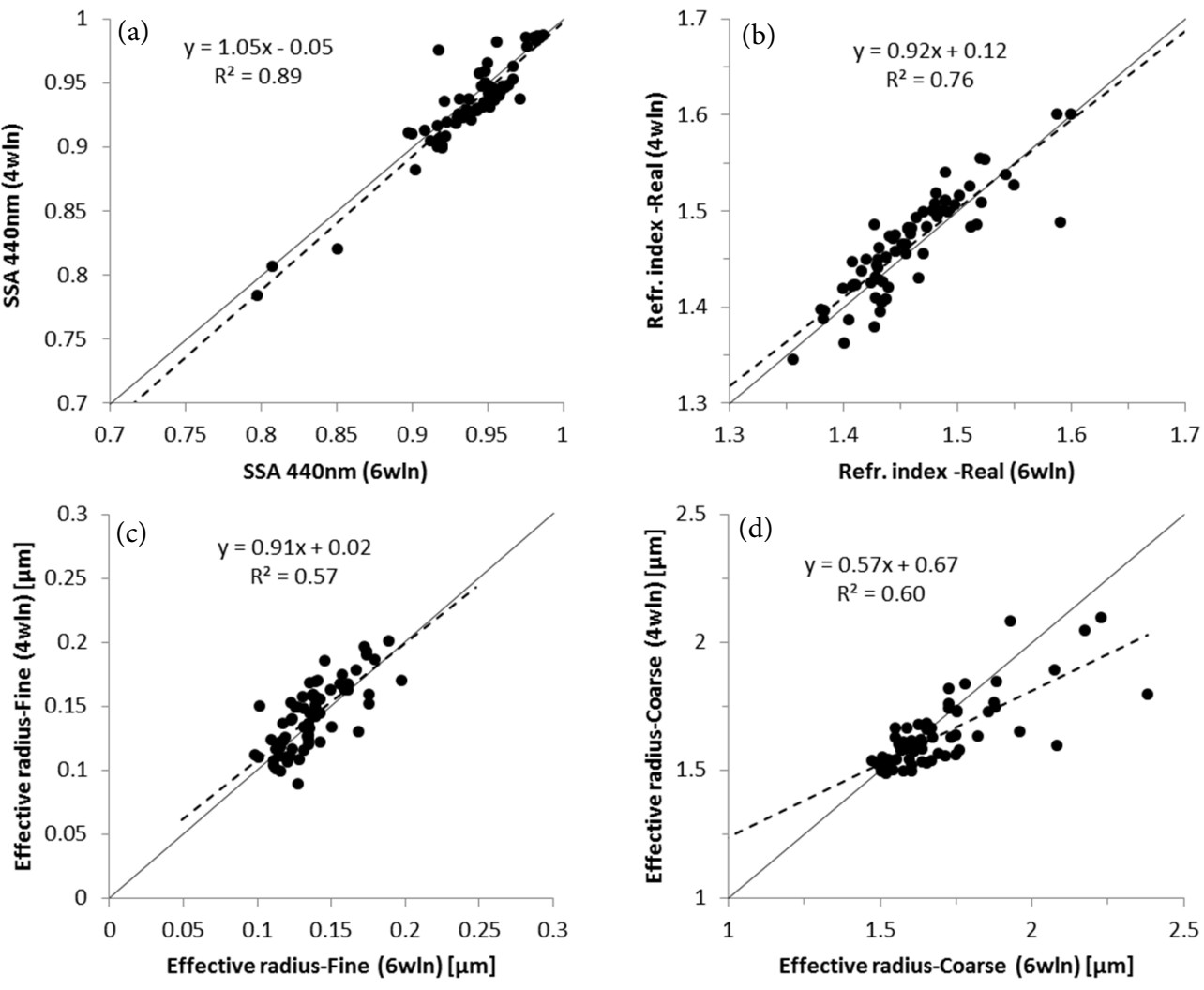

**Figure 6.** Comparison of inversion retrievals using 4 spectral channels (440, 675, 870, 1020nm) or 6 channels (440, 500, 675, 870, 1020 and 1640nm) in the diffuse sky radiance: (a) Single scattering albedo (440nm); (b) Real part of the refractive index; (c) Effective radius of the fine mode; (d) Effective radius of the coarse mode. Data from Cimel #789 at 'Barbados_Saltrace' using almucantar and principal plane observations. Solid lines indicate the 1:1 line. Dashed lines indicate the linear fit to the displayed data, for which the line equation and correlation coefficient are also provided.

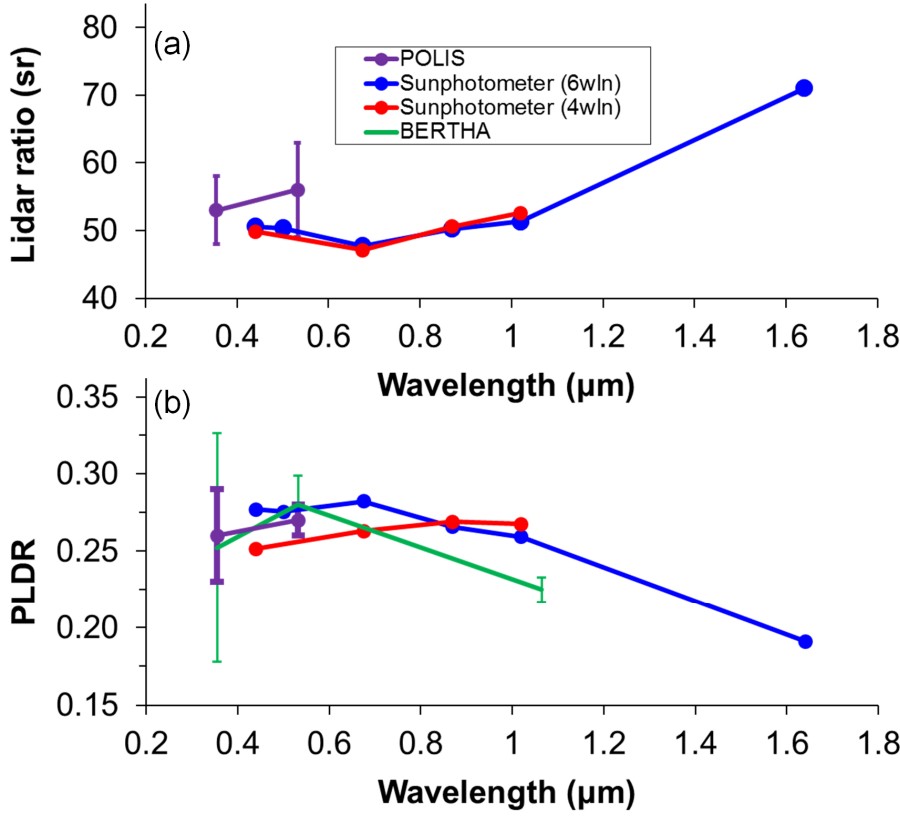

**Figure 7.** Wavelength dependence of (a) lidar ratio (LR); (b) Particle linear depolarization ratio (PLDR), for dust cases during SALTRACE. Data from POLIS lidar (Groß et al., 2015) and BERTHA lidar (Haarig et al., 2017) are compared to the 4-wavelength and 6-wavelength sun photometer retrievals. Vertical bars in lidar data indicate uncertainty due to systematic errors.

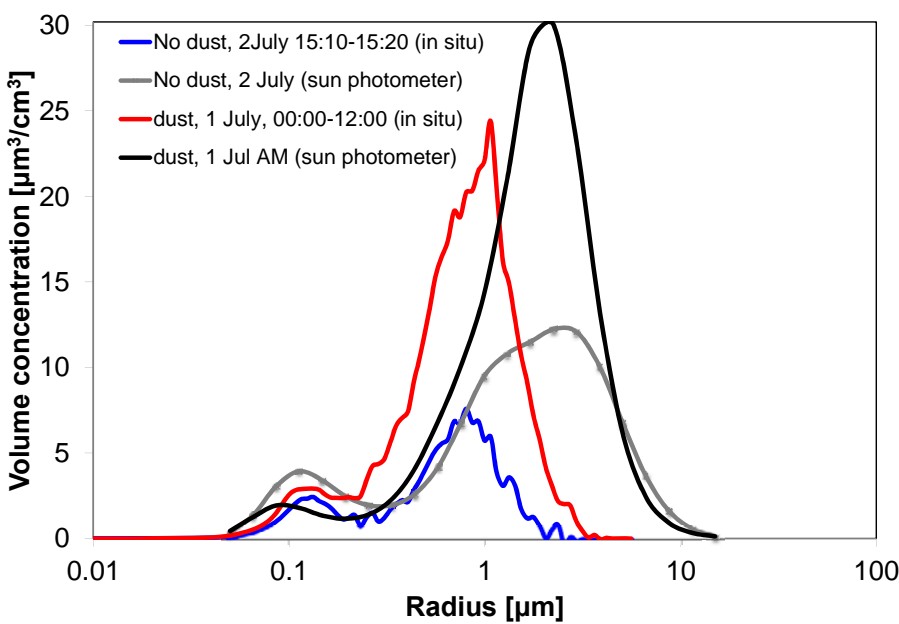

**Figure 8.** Volume particle size distributions for 1st July 2013 (dust case) and 2nd July 2013 (no-dust case) measured in-situ and by inversion of sun photometer data. The latter have been converted to concentration using a layer height derived from the co-located lidar observations (see details in the text).

**Table 1.** Sun photometer instruments deployed at CIMH and Ragged Point sites during SALTRACE. The instrument type, location, spectral range, observation period, owner and the sampling interval of different measurement types are indicated.

| Instrument | Location | Obs. period | Spectral range | Direct Sun (AOD) | Sky radiance | Polarization | Owner |
|---|---|---|---|---|---|---|---|
| Cimel #789 | CIMH | 7 Jun-11 Jul, 2013 | 340-1640nm | 3min | 1h | — | UVa |
| Cimel #440 | CIMH | 12 Jun 2013,14 Jul 2014 | 340-1640nm | 15min | 1h | 1h | TROPOS |
| SSARA-P | CIMH | 10 Jun-11 Jul, 2013 | 340-1550nm | 2sec | 30min | 30min | LMU |
| Microtops | CIMH | 7 Jun-14 Jul, 2013 | 305-1020nm | manual (daily) | — | — | LMU |
| Cimel #305 | Ragged Point | Continuous | 340-1020nm | 3min | 1h | — | U. Miami |

**Table 2.** Statistics of sun photometer observations during SALTRACE campaign during the dust episodes: aerosol optical depth (AOD), Ångström Exponent, precipitable water, single scattering albedo (SSA), Real and Imaginary part of the refractive index, volume concentrations (VolCon) of the total size distribution (T) and the coarse mode (C), fine mode fraction (FMF) of the size distribution, effective radius (EffR) of the total size distribution and the coarse mode, fraction of spherical particles, lidar ratio and particle linear depolarization ratio (PLDR).

| | Mean $\pm$ Std.Dev | Median | 5th Perc. | 95th Perc. |
|---|---|---|---|---|
| AOD (340nm) | 0.284± 0.123 | 0.287 | 0.088 | 0.483 |
| AOD (500nm) | 0.262± 0.125 | 0.266 | 0.066 | 0.464 |
| AOD (1640nm) | 0.200± 0.104 | 0.203 | 0.043 | 0.367 |
| Ångström Exp. | 0.15 ±0.12 | 0.11 | 0.04 | 0.39 |
| Water [cm] | 3.53±0.69 | 3.53 | 2.44 | 4.73 |
| SSA (440nm) | 0.942± 0.035 | 0.937 | 0.900 | 0.986 |
| SSA (1020nm) | 0.979± 0.017 | 0.984 | 0.944 | 0.993 |
| Refr-Real(440nm) | 1.474±0.044 | 1.475 | 1.415 | 1.544 |
| Refr-Imag.(440nm) | 0.003±0.002 | 0.003 | 0.001 | 0.005 |
| VolCon(T) [$\mu m^3/\mu m^2$] | 0.199±0.067 | 0.180 | 0.132 | 0.300 |
| VolCon(C) [$\mu m^3/\mu m^2$] | 0.184±0.064 | 0.166 | 0.123 | 0.286 |
| FMF | 0.088±0.028 | 0.081 | 0.051 | 0.138 |
| EffR-T [$\mu m$] | 0.912±0.180 | 0.942 | 0.575 | 1.169 |
| EffR-C [$\mu m$] | 1.615±0.121 | 1.586 | 1.495 | 1.838 |
| Sphericity [%] | 23±28 | 12 | 0.1 | 82 |
| Lidar ratio (440nm) [sr] | 50±7 | 49 | 39 | 61 |
| Lidar ratio (1020nm) [sr] | 53±9 | 54 | 39 | 68 |
| PLDR (440nm) | 0.25±0.06 | 0.28 | 0.13 | 0.31 |
| PLDR (1020nm) | 0.27±0.05 | 0.29 | 0.16 | 0.32 |