# Peer review of "Sun photometer retrievals of Saharan dust properties over Barbados during SALTRACE"

_Atmospheric Chemistry and Physics, 2019_

## Referee Comment (RC1) · Anonymous Referee #1 · 8 Jul 2019

The study from C. Toledano et al. represents a fair analysis of the sunphotometer data retrieved during the Saharan Aerosol Long-range Transport and Aerosol-Cloud-Interaction Experiment (SALTRACE) campaign held at Barbados in the Caribbean during June-July 2013. This analysis provides useful information for the comparison and itnerpretation with other colocated vertical and insitu measurements, but it also includes some analysis exercises that broaden the basic analysis of retrievals provided by AERONET, for a deeper insight on the aerosol properties. The techniques and methods are well known and described in many other previous references. The study is considered adequate for this journal. The article is well written, although it would benefit of a final revision by a native English speaker. However, I am not able to find

important flaws on the English grammar or style.

General comments:

- The SSARA instrument is introduced in the abstract and briefly described in the introduction. Some comments about the calibration are included in the instrumentation section too. However, its data (AOD) is only represented in Figure 1 for comparison with the other instruments AOD, without proper discussion in the text body. I expected more analysis on this instrument after being included in the abstract. Therefore, expand the SSARA analysis please. Otherwise, it could be avoided in the abstract.

- The analysis of the aerosol retrievals when extending the inversion to 6 wavelengths (i.e. including 500 and 1640 nm) is an interesting section in the study. I wonder if the same conclusions were/would be reached if only i.e. 1640 nm channel is added, as the 500 nm does not really extend the interval. Can the authors comment their findings?

Specific comments:

- Page 4, line 7: Correct "Methodology" - Page 4, line 13: any reference for the analysis of uncertainty of the AOD obtained with SSARA-P? I expect 0.01-0.02 to be the uncertainty of field Cimel instruments, but if the SSARA is calibrated by a standard Langley plot at a high site, then I would expect a lower uncertainty on AOD. - Page 4, line 19: what the 0.003 difference between version 2 and version 3 refers to? Is it the difference found between both datasets from the campaign? Does it represent the RMSD? Please state in the text. - Page 5, line 10: correct "specialissue" - Page 5, lines 17-22: could you give the numerical value of the average AOD found during the episodes? - Table 2: these results are obtained for the aerosol properties during the dust episodes, as it is said in the text (page 5, line 31). But it would be good to add a note in the table caption. - Page 7, line 11-12: why the SSA is believed to be smaller in Morroco? Was it related to a higher pollution? - Page 8, line 14: correct "unfortunately" - Page 8, line 16: LPDR or PLDR? Use the same everywhere. - Page 8, line 21: "is too high for the sunphotometer". This sentence seems ambiguous to me. Please rewrite. - Page 8,

line 23: correct "particular" - Page 8, line 27: Correct "The the" - Page 9, line 3: Please include references, as comparisons between columnar inversions and insitu profiles for dust cases already exist (see for example www.atmos-chem-phys.net/15/8479/2015/) - Figure 8 and related analysis: a broad estimation of more comparable volume distributions could be performed by assuming that the dust layer is distributed evenly in a layer. If this assumption is valid for this campaign day (supported by vertical measurements) it would be interesting to see a modified plot with both distributions in units um3cm-3. - Page 9: I miss results of SSARA from radiance measurements. - Figure 2: This plot does not seem to be a log-log plot as stated in the caption but a semi-log plot. Please check and comment accordingly for the related discussion. - Figure 2: Given that AOD at 2um is not a experimental but extrapolated value, in my opinion it should be better represented with a different shape to avoid confusing the reader (even if highlighted with an external circle).
* * *

---

## Referee Comment (RC2) · Anonymous Referee #2 · 15 Jul 2019

Comment on "Sun Photometer retrievals of Saharan dust properties over Barbados during SALTRACE" by Carlos Todelano et al

The SALTRACE measurement campaign presents a significant opportunity to characterize dust properties before and after trans-Atlantic transport. In this paper, Todelano et al combine sun photometer and lidar measurements from ground-based platforms in Barbados with established inversion techniques to determine aged dust properties. The analysis is extensive and contains many useful measurements that can be used for climate model validation and comparison with future measurements. I also appreciate the many useful comparisons with previous campaigns such as SAMUM which put

the measurements in context.

The manuscript would benefit from (many) grammatical corrections but these do not impact the overall quality of the analysis and the paper deserves to be published in ACP after a few minor (mostly grammatical) issues are dealt with. I also acknowledge and agree with the General Comments from Reviewer 1.

Specific comments [P1 L10] "The sun photometer ... was used in the retrieval to investigate possible improvements " – add "to aerosol size retrievals2=" [P1 L13] "However the comparison of size distributions" – comparison -> differences [P2 L8] Remove "so called" from the sentence [P2 L9] "can only be tackled with a combination of long-term observations of key variables using ground-based, airborne and satellite techniques" – or similar amendment [P2 L19] Reword the last sentence of paragraph 3 – it currently reads that the AERONET data resulted in typical dust conditions during SALTRACE, rather than it demonstrating that typical dust conditions were observed [P3 L7] Sentence ending "relate them to the co-located measurements" – co-located measurements of what? [P4 L13] "Similar uncertainty is found for SSARA-P" provide a suitable reference or evidence [P4 L19] Grammatical change – "The use of version 2 AOD is needed" should be "The use of version 2 is chosen" or selected [P5 L22] "Moreover, all the instruments co-located at CIMH agree within the nominal AOD uncertainty (0.02)" – This statement does not seem to be true on a day-to-day basis from looking at Fig. 1a. For instance on the 29/30th June SSARA-P suggests AOD of 0.15 with the Cimel measurements much above this [P6 L2] "We used the 1% percentile of AOD within each month" – why did you chose this rather arbitrary value? What happens if you select the 5% percentile etc. The AOD threshold of 0.04 does not agree with the 0.2 threshold you use to for Table 2 – I don't understand why you used two different thresholds [P6 L22] Sentence beginning "The AE of dust seems to be lower in SAMUM-2 and SALTRACE than SAMUM-1" – This is my only real qualm with the methodology – the failure to delineate successfully between the different forms of aerosol present during the observation period. The authors use a tenuous threshold of AOD = 0.15

(again different to the previous thresholds of 0.2 and 0.04) to delineate marine from dust aerosol, but ultimately there will be some marine aerosol present in the dust retrievals. This should perhaps be added as a caveat here and in the conclusions – that the measurements in Table 2 represent a mixture of dust (pre-dominant) with some marine aerosol contamination [P8 L31] "get facilitated" -> "become similar"

---

## Author Comment (AC1) · 27 Sep 2019

General comments:

- The SSARA instrument is introduced in the abstract and briefly described in the introduction. Some comments about the calibration are included in the instrumentation section too. However, its data (AOD) is only represented in Figure 1 for comparison with the other instruments AOD, without proper discussion in the text body. I expected more analysis on this instrument after being included in the abstract. Therefore, expand the SSARA analysis please. Otherwise, it could be avoided in the abstract.

Only limited results (basically AOD) could be retrieved with SSARA, because some pointing problems in the almucantars prevent from using the sky radiances in the inversion. Therefore no further results can be shown for this instrument. We have added some specific comments to the differences between AOD data of SSARA and the Cimels in section 4.1, also in response to Referee.#2.

- The analysis of the aerosol retrievals when extending the inversion to 6 wavelengths (i.e. including 500 and 1640 nm) is an interesting section in the study. I wonder if the same conclusions were/would be reached if only i.e. 1640 nm channel is added, as the 500 nm does not really extend the interval. Can the authors comment their findings?

Yes, the use of 500nm only increases the robustness of the retrieval, since it adds basically redundant information. The additional information (mainly about the coarse mode) is given by the 1640nm channel. A footnote has been added for clarification: "The key additional channel is 1640nm; the 500nm channel may increase robustness of the results but would not produce the observed changes in the coarse mode retrieval."

Specific comments:
- Page 4, line 7: Correct "Methodology"
Done
- Page 4, line 13: any reference for the analysis of uncertainty of the AOD obtained with SSARA-P? I expect 0.01-0.02 to be the uncertainty of field Cimel instruments, but if the SSARA is calibrated by a standard Langley plot at a high site, then I would expect a lower uncertainty on AOD.

This is true if sufficient number of Langley calibrations can be performed (see for instance the uncertainty discussion in Toledano et al., ACP 2018, https://doi.org/10.5194/acp-18-14555-2018). We averaged only 5 morning Langleys and therefore an uncertainty of 0.6% to 1% depending on wavelength is to be expected, about 0.006-0.01 in AOD. As noted by the reviewer, this is slightly better than the nominal uncertainty for field Cimel instruments. But the SSARA has a front window that is much more exposed to dust than the Cimels, and we prefer being conservative with the uncertainty estimation, based on the small AOD changes observed after the daily cleaning of the SSARA front window.

- Page 4, line 19: what the 0.003 difference between version 2 and version 3 refers to? Is it the difference found between both datasets from the campaign? Does it represent the RMSD? Please state in the text.

This is the difference found between both datasets from the campaign period. It is produced by slightly different AOD retrieval algorithm (mainly due to temperature corrections and gas absorptions that are incorporated in version 3). Other than that, it's the same raw values and nearly identical algorithm (for airmass computation, Rayleigh corrections, etc.), so the results are nearly identical for each single data point.

- Page 5, line 10: correct "specialissue"
Done

- Page 5, lines 17-22: could you give the numerical value of the average AOD found during the episodes?

The average AOD (500nm) for each of the episodes indicated in Figure 1 is:
10-20 Jun:      0.42
22-24 Jun:      0.24
26 Jun-1 Jul:   0.29
4-6 Jul:        0.22
8-13 Jul:       0.35
We have added in the text: "Mean AOD values for the events ranged between 0.22 and 0.42."
-Table 2: these results are obtained for the aerosol properties during the dust episodes, as it is said in the text (page 5, line 31). But it would be good to add a note in the table caption.
Done.

- Page 7, line 11-12: why the SSA is believed to be smaller in Morroco? Was it related to a higher pollution?
That's a possible explanation. In Cape Verde and Barbados the background aerosol is pure marine and local pollution can be neglected in comparison with the atmospheric column and the Saharan Air Layer. But maybe pollution could affect the measurements in Morocco. Data analysis in SAMUM-1 indicated the presence of soot particles in the fine mode (Schladitz et al, Tellus 2008).
- Page 8, line 14: correct "unfortunately"
Done.
- Page 8, line 16: LPDR or PLDR? Use the same everywhere.
Done.

- Page 8, line 21: "is too high for the sunphotometer". This sentence seems ambiguous to me. Please rewrite.
Changed to: "[…] even though the PLDR at 1020-1064nm derived from the sun photometer is in any case higher than indicated by the lidar uncertainty estimates."
- Page 8, line 23: correct "particular"
Done.
- Page 8, line 27: Correct "The the"
Done
- Page 9, line 3: Please include references, as comparisons between columnar inversions and insitu profiles for dust cases already exist (see for example www.atmos-chem-phys.net/15/8479/2015/)
Good point, thank you. Some references have been added to give a broader picture of this kind of comparison.

- Figure 8 and related analysis: a broad estimation of more comparable volume distributions could be performed by assuming that the dust layer is distributed evenly in a layer. If this assumption is valid for this campaign day (supported by vertical measurements) it would be interesting to see a modified plot with both distributions in units um3cm-3.
We have followed the suggestion and changed the figure 8 because we agree that applying this assumption is a more adequate way to compare the in-situ and sun photometer data. The sun photometer data are compared to concentration in um3/cm3 using an approximate scale height deduced from the lidar layering analysis presented in Gross et al., 2015. Anyway, the qualitative comparison presented in the submitted manuscript was already very discouraging because the inlet cutoff prevents from any meaningful comparison in the coarse mode. Thus a quantitative comparison (that in addition would require robust uncertainty estimation) is not intended at all here.
We added the following text:
"We have converted the column size distribution from the sun photometer (originally in um3/um2) to concentration in um3/cm3 assuming that the dust layer is distributed

evenly in a layer and using a scale height of 4 km for the dust case and 1.5 km for the marine case (data from the co-located POLIS lidar, Groß et al., 2015, Fig. 1).

- Page 9: I miss results of SSARA from radiance measurements.
As explained in the general comments, only limited results (basically AOD) could be retrieved with SSARA, because some pointing problems in the almucantars prevent from using the sky radiances in the inversion. Therefore no further results can be shown for this instrument.

- Figure 2: This plot does not seem to be a log-log plot as stated in the caption but a semi-log plot. Please check and comment accordingly for the related discussion.
It is log-log (apparently the Y axis doesn't look like, but it is).

- Figure 2: Given that AOD at 2um is not a experimental but extrapolated value, in my opinion it should be better represented with a different shape to avoid confusing the reader (even if highlighted with an external circle).
Done.

**Anonymous Referee #2**

Specific comments

[P1 L10] "The sun photometer ... was used in the retrieval to investigate possible improvements " – add "to aerosol size retrievals2="
Done

[P1 L13] "However the comparison of size distributions" – comparison -> differences
Done

[P2 L8] Remove "so called" from the sentence
Done

[P2 L9] "can only be tackled with a combination of long-term observations of key variables using ground-based, airborne and satellite techniques" –or similar amendment
Done

[P2 L19] Reword the last sentence of paragraph 3 – it currently reads that the AERONET data resulted in typical dust conditions during SALTRACE, rather than it demonstrating that typical dust conditions were observed
Yes indeed! We have rewritten the sentence.

[P3 L7] Sentence ending "relate them to the co-located measurements" – co-located measurements of what?
Co-located aerosol measurements. Done.

[P4 L13] "Similar uncertainty is found for SSARA-P" provide a suitable reference or evidence
Done.

[P4 L19] Grammatical change – "The use of version 2 AOD is needed" should be "The use of version 2 is chosen" or selected

Done.

[P5 L22] "Moreover, all the instruments co-located at CIMH agree within the nominal AOD uncertainty (0.02)" – This statement does not seem to be true on a day-to-day basis from looking at Fig. 1a. For instance on the 29/30th June SSARA-P suggests AOD of 0.15 with the Cimel measurements much above this
We have added "for simultaneous measurements." The figure 1a spans over 5 weeks and data are too close to allow distinguishing few minutes time difference. As example we show a zoom of some days. On 27 June, that had almost clear sky, the AOD differences between the 2 cimels and the SSARA are <0.02 (note that data in red are from another location, Ragged Point). On 28 June, it was a cloudy day and many data were removed in the cloud-screening process. The difference in the measurement acquisition (3 single data in 1 minute for Cimels, 2 sec sampling for SSARA resulting in 30 observations per minute), may lead to different results in the cloud-screening process. As a consequence of the different rain sensors used in each instrument each photometer might resume measurements differently in case of showers.
Finally, the last few measurements of SSARA on the 28 June (AOD about 0.15) occur 6 minutes later than the last Cimel measurement. It could be that some dust plume (or cirrus) produced this quick change.

[Figure]

In any case, for clear and stable conditions, simultaneous AOD measurements from SSARA and Cimel were the same within the estimated uncertainties. The following clarification was added in the AOD section:
"Some minor differences between SSARA and the AERONET Cimels are to be expected in cloudy days (e.g. 17-Jun, 28-Jun). This is due to the different data sampling of the instruments ('triplets' or 3 measurements within 1 minute for the Cimels; 2 second sampling for the SSARA), that may yield non-simultaneous data as well as different results in the cloud-screening process."

[P6 L2] "We used the 1% percentile of AOD within each month" – why did you chose this rather arbitrary value? What happens if you select the 5% percentile etc. The AOD threshold of 0.04 does not agree with the 0.2 threshold you use to for Table 2 – I don't understand why you used two different thresholds
We simple wanted to avoid using the minimum value in the period to evaluate the background AOD (it is more subject to errors). The 5$^{th}$ percentile is similar although the baseline we try to identify (the cleanest conditions) are closer to percentile 1. It would not change any conclusion. The boundaries for aerosol type predominance are always somewhat arbitrary and mixtures are always present in column-integrated variables like AOD. See also answers below.
The AOD>0.2 threshold in Table 2 refers to the minimum AOD considered to ensure the quality of the inversion retrieval. Magnitudes like single scattering albedo cannot be properly estimated with low AOD (there is not enough aerosol signature in the sky radiance). The AOD>0.15 allows the (inevitably tenuous, this is true) separation of dust-dominated vs. marine-dominated scenes. But if we need to look at optical

properties derived from inversion (complex refractive index, SSA, etc.), then we must restrict to AOD>0.2 to keep uncertainties low.

[P6 L22] Sentence beginning "The AE of dust seems to be lower in SAMUM-2 and SALTRACE than SAMUM-1" – This is my only real qualm with the methodology – the failure to delineate successfully between the different forms of aerosol present during the observation period. The authors use a tenuous threshold of AOD = 0.15 (again different to the previous thresholds of 0.2 and 0.04) to delineate marine from dust aerosol, but ultimately there will be some marine aerosol present in the dust retrievals. This should perhaps be added as a caveat here and in the conclusions – that the measurements in Table 2 represent a mixture of dust (pre-dominant) with some marine aerosol contamination

The discussion about the background aerosol intended to show that there is always marine aerosol present, therefore a mixture (in the column) of dust and marine. For AOD>0.15 we can expect a clear predominance of dust, so that the analyzed properties are very close to those of dust, with only a minor marine contribution. A clarification has been added to the text: "
[revised manuscript text omitted]

| AOD (500nm) | 0.262$\pm$ 0.125 | 0.266 | 0.066 | 0.464 |
| AOD (1640nm) | 0.200$\pm$ 0.104 | 0.203 | 0.043 | 0.367 |
| Ångström Exp. | 0.15 $\pm$0.12 | 0.11 | 0.04 | 0.39 |
| Water [cm] | 3.53$\pm$0.69 | 3.53 | 2.44 | 4.73 |
| SSA (440nm) | 0.942$\pm$ 0.035 | 0.937 | 0.900 | 0.986 |
| SSA (1020nm) | 0.979$\pm$ 0.017 | 0.984 | 0.944 | 0.993 |
| Refr-Real(440nm) | 1.474$\pm$0.044 | 1.475 | 1.415 | 1.544 |
| Refr-Imag.(440nm) | 0.003$\pm$0.002 | 0.003 | 0.001 | 0.005 |
| VolCon(T) [$\mu m^3/\mu m^2$] | 0.199$\pm$0.067 | 0.180 | 0.132 | 0.300 |
| VolCon(C) [$\mu m^3/\mu m^2$] | 0.184$\pm$0.064 | 0.166 | 0.123 | 0.286 |
| FMF | 0.088$\pm$0.028 | 0.081 | 0.051 | 0.138 |
| EffR-T [$\mu m$] | 0.912$\pm$0.180 | 0.942 | 0.575 | 1.169 |
| EffR-C [$\mu m$] | 1.615$\pm$0.121 | 1.586 | 1.495 | 1.838 |
| Sphericity [%] | 23$\pm$28 | 12 | 0.1 | 82 |
| Lidar ratio (440nm) [sr] | 50$\pm$7 | 49 | 39 | 61 |
| Lidar ratio (1020nm) [sr] | 53$\pm$9 | 54 | 39 | 68 |
| PLDR (440nm) | 0.25$\pm$0.06 | 0.28 | 0.13 | 0.31 |
| PLDR (1020nm) | 0.27$\pm$0.05 | 0.29 | 0.16 | 0.32 |

---

## Author Response (AR2)

Dear co-editor,

Please find below the response to your comments. The corresponding changes have been added to the manuscript. Many thanks for your clarifications.

Sincerely,

Carlos Toledano and co-authors

**Co-Editor Decision: Publish subject to minor revisions (review by editor)** (01 Oct 2019) by Claire Ryder
Comments to the Author:
Please make the following changes/responses to the manuscript:

1) The response to reviewer 2, in their point concerning the first percentile of AOD, for establishing the marine background, is still not clear. Please clearly explain why this value (0.04 AOD for June/July) is necessary in addition to the 0.15 AOD threshold for dust/marine aerosol.

The important point was to highlight that the background aerosol is marine, and that there is always a contribution of marine aerosol particles to the atmospheric column (AOD). The sentence "Whenever AOD was above this level [0.04], the presence of dust in the atmospheric column was considered." was misleading and has been deleted. New sentence is: "This background marine aerosol always contributes to the aerosol optical depth observed in the atmospheric column. " The 0.15 threshold ensures the predominance of dust and it is the value that has been considered for type separation in the study.

2) Please include more information on the in-situ measurements. Simply citing other references is not sufficient. Since data from them is presented in Figure 8, the specific instruments used in this case should be detailed and size ranges and any relevant processing described. Relevant altitude ranges or flight manoeuvers (e.g. constant altitude flight legs) used to gather the data should be described, since this may impact the comparison to sunphotometer data. Information on the inlet and size measurement limitations should be described, since this has a large impact on the data shown in Figure 8.

We have added a short description of the POLIS and BERTHA lidars, as well as the ground-based in situ size distribution instrumentation at the end of section 2 (sites and instrumentation).

3) p9 l30 - "The relevant comparison here is the shape of the size distributions" - if this is the case, the size distributions should be normalized so that the true shape differences can be compared.

Reviewer #2 asked us to change figure 8 from normalized size distributions (that was our original fig.8). As explained in the response to reviewers, we now converted the column to in-situ using height information from the lidar. This allows quantitative comparison apart from the shape comparison. We think this approach is better because it has more information for comparison. To avoid confusion, the sentence about the shape has been reformulated: "About the shape of the size distributions,…"

4) Fig 5 - caption should name instruments used.
Done.

5) Fig 6 caption - should state what solid and dashed lines indicate.
Done.

6) p9 l7 - a better wording may be, "Only in very well mixed atmospheres with a single predominant aerosol type is it possible to tackle this kind of comparison."
Thanks!

7) p9 l22-23 - Table 2 only appears to show sunphotometer measurements. Please clarify this statement to indicate where the in-situ SSA values are shown/taken from.
All data in Table 2 are derived from sun photometer, including the SSA values and lidar parameters, which are all provided by the Dubovik inversion of AOD+sky radiances.

8) p9 l27-29 - a scale height implies a decrease in concentration with altitude. (not distributed evenly).

This is true. We have changed the text to "layer height". The calculation is just approximate; no accurate quantitative comparison is intended. Therefore we used an evenly distributed layer for simplicity.

9) p10 l29-30 - "The closure approach, aiming at the characterization of the same aerosol parameters with various independent methods, is the next step to be carried out with the SALTRACE dataset." - please make this sentence less vague and clearer.

[revised manuscript text omitted]